# RobARtics: Using Augmented Reality to enhance Robotics Competitions

BERNARDO MARQUES, DigiMedia, DeCA; IEETA, DETI, LASI, University of Aveiro, Portugal

JOÃO ALVES, IEETA, DETI, LASI, University of Aveiro, Portugal

EURICO PEDROSA, IEETA, DETI, LASI, University of Aveiro, Portugal

RENATO CABRAL, IEETA, DETI, University of Aveiro, Portugal

SAMUEL SILVA, IEETA, DETI, LASI, University of Aveiro, Portugal

BEATRIZ SOUSA SANTOS, IEETA, DETI, LASI, University of Aveiro, Portugal

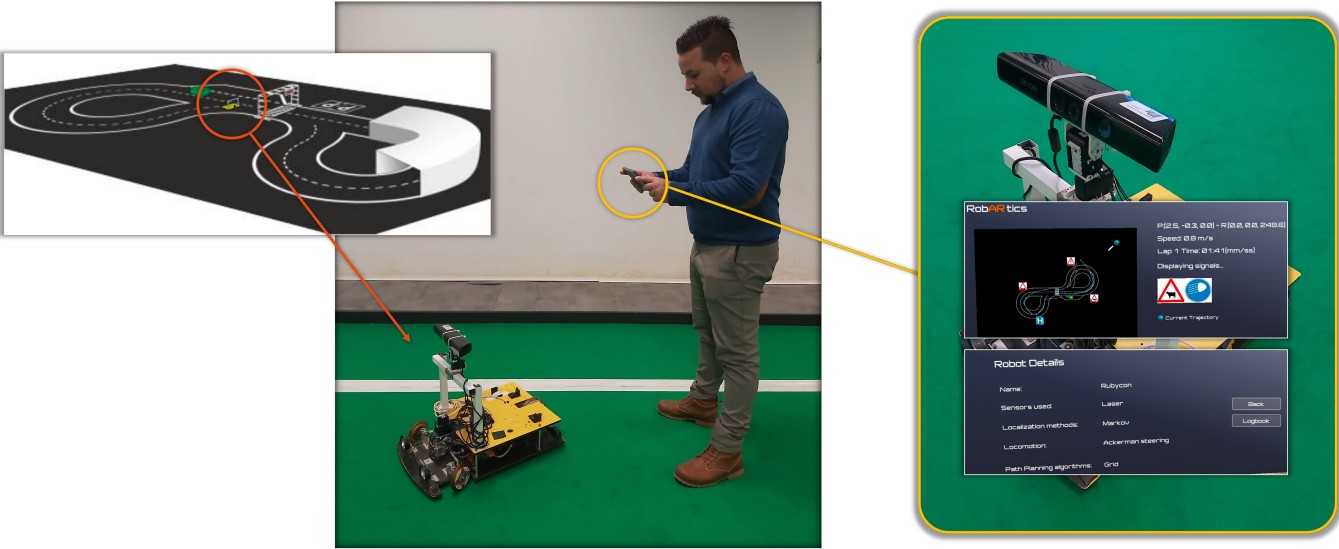

Fig. 1. Scenario having a researcher using the robARtics tool during a robotics competition to obtain additional information regarding the robot behaviour in order to conduct a more informed decision.

The technological advancements in recent years have allowed an increase in both the robot's hardware and software capabilities. In this context, robotic competitions have been explored, allowing research teams to participate in a series of challenges, generally supervised by referees, and with large numbers of public attendance. However, during such competitions, there is a lack of methods to provide information to everyone involved, since attendees often don't get the chance of getting closer to the robots, and researchers often need to interrupt the competition and connect additional hardware to learn more about what was happening with the robot. This work explores a new method of amplifying what distinct audiences are able to see during such competitions, utilizing a mobile Augmented Reality (AR) tool, aimed at providing users with an effective way of obtaining more information about the robots and robotics competitions. This tool was designed using a Human-Centered Design (HCD) approach, having the collaboration of target users and domain experts during its design and development. This participatory process allowed to iteratively test the various phases of the tool proposed and integrate the feedback collected in each one.

*Proceedings of HRI-23 Workshop on Virtual, Augmented, and Mixed Reality for Human-Robot Interaction, March, 2023, Stockholm, SE*

CCS Concepts: • **Human-centered computing → Mixed / Augmented Reality**; • **Computer Systems oOganization → External Interfaces for Robotics**; • **Computing Methodologies → Robotic Planning**.

Additional Key Words and Phrases: Robotics Competition, Human-Robot Interaction, Augmented Reality, Human-Centered Design

**ACM Reference Format:**
Bernardo Marques, João Alves, Eurico Pedrosa, Renato Cabral, Samuel Silva, and Beatriz Sousa Santos. 2023. RobARtics: Using Augmented Reality to enhance Robotics Competitions. In *Proceedings of HRI-23 Workshop on Virtual, Augmented, and Mixed Reality for Human-Robot Interaction (VAM-HRI), March, 2018, Stockholm, SE.* ACM, New York, NY, USA, 7 pages.

## 1 INTRODUCTION

Augmented Reality (AR) technologies have been explored over the latest decades with applications in various fields (e.g., industry, healthcare, education, entertainment, serious games, among others) [3–5, 7, 11, 17, 18]. In recent years, other relevant areas of application have deserved further attention by the AR research community as is the case of Human-to-Human Co-located and Remote Collaboration [14, 15], as well as Human-Robot Interaction (HRI) or Human–Robot Collaboration (HRC) thanks to the advancements in recent hardware and software solutions [1, 9].

Solutions exploring AR have the potential to improve decision-making and interaction with robots by enhancing humans awareness and spatial understanding through visualization of situated real-time responsive computer-generated information that is superimposed over the real-world environment [12, 13, 16].

Although AR has been used for HRI in various scenarios of application, most of them are in the industry sector (e.g., safety, assembly, quality assurance, welding, painting, and many others) [2, 6, 8, 10, 19, 20]. In this vein, a scenario that has not been explored yet is assisting humans during robotic competitions. These competitions can be described as events that consist of multiple tasks that must be performed by robots, usually competing to best each other during the competition. Such competitions have come a long way since their inception dating back to the 1970s.

An example is the *Festival Nacional de Robótica* (FNR)[1], an event designed to "promote Science and Technology to researchers, students and general public through the use of automated robots", promoted by the *Sociedade Portuguesa de Robótica* (SRP). Alongside this, it is also where many Portuguese and international teams are accepted towards the RoboCup, the international robotic competition. The FNR is also the home of the International Conference on Autonomous Robot Systems and Competitions (ICARSC), an event where researchers from many parts of the world present their latest advances in robotics.

Given that multiple competitions exist within the scope of the FNR, to clarify, in this work, we have focused on the autonomous driving challenge. The intention is to have autonomous robots capable of traversing a closed circuit completing two laps around a track (with some similarities to a regular traffic road) in the shortest time possible while accruing the least amount of penalties possible. The track consists of an 8-pattern track, with some variations including zebra crossings, tunnels, or traffic lights, none of which are previously known to the robot (Figure 2-2).

The challenge is divided into smaller tasks, that are performed on 3 consecutive days. There are driving challenges, parking challenges, and vertical sign detection challenges [2]. To further elaborate, during the competition, there are available twelve unique road signs, that are placed on the right side of the track. These are equally grouped in triangular warning signs, round mandatory signs, and square-shaped service signs (Figure 2-1).

Regarding the robot used by our university team, it is designated as ROTA, a tricycle composed of two wheels with directional control, but without traction and a directionless third wheel with traction behind, adopting the Ackerman steering model (Figure 2-4). Its high-level design architecture was developed using Robot Operating System (ROS), a framework designed to establish communication between engineering components. The control system for this robot is composed of a high-level layer, connected to a computer, which is responsible for coordinating the robot's movement. This layer is linked to a low-level layer through Universal Serial Bus (USB). The low-level layer itself is composed of a Controller Area Network (CAN) of micro-controllers. On top of it, it has a Kinect camera, equipped with RGB-D and a Laser Range Finder (LRF). The RGB-D

component allows sign identification, as well as capturing images while the robot is in motion. The LRF is used to detect and avoid obstacles.

Although the technology used in these events is constantly being upgraded, sometimes there's uncertainty when it comes to the challenge's outcomes. All things considered, this type of competition implies a huge amount of data that needs to be analyzed and compared, for instance, to understand how different robot configurations influence its performance. To have a grasp of the differences among multiple runs, it is paramount to understand how the robot trajectories or speed were influenced, for particular segments of the route, and infer how to further adjust the configurations. While this kind of analysis might be performed using unsupervised computational methods, the proposal of visual methods for the exploration of the data, in place, fosters new insights and can, in the future, inform the proposal of such methods. In addition, disagreements between the competition staff and the participants may arise in ambiguous situations that can have outcomes that are hard to tell, such as deciding which robot crossed the finish line first during a race. Likewise, with the rising interest of the general public in attending robot competitions, there is an urge to provide a contextualization of what is happening, particularly regarding robot performance. For instance, enabling an understanding of which sign the robot just recognized, which direction it took at a track bifurcation, or where the robot went out of the track.

To address these challenges, in this work, we propose the use of AR to assist distinct types of audiences during robot competitions, in particular, allowing exploration, and analyzing relevant data from a given robot. A Human-Centered Design (HCD) methodology was used to understand the context, i.e., scenarios, personas, motivations, and challenges, as well as define requirements and identify relevant features in which AR could provide a step forward in understanding what happens with a robot during a competition.

The remnant of this paper is structured as follows. Section 2 describes the methodology used for assisting robotics competition while using AR. Then, Section 3 describes the robARtics tool, including its features according to the user profile, as well as the technologies used to implement it. Plus, initial results from a preliminary user study are described, as well as insights from domain experts in the competition selected as use-case. Last, conclusions and future research directions are drawn in Section 4.

## 2 METHODOLOGY FOR ASSISTING IN ROBOTICS COMPETITION

Next, the scenarios and personas that define the goals of this research are introduced. A HCD methodology was followed and as a first step to better understand the target scenario and audience, this work started with informal interviews with potential users followed by brainstorming sessions involving software engineers and augmented reality and robotics experts. Several stakeholders were identified, along with their main motivations, and an analysis of the current practices around robotic competitions lead to multiple scenarios, which were modified to encompass the use of a novel set of supporting features. These were materialized in user profiles, adopting some of the concepts related to personas (e.g., explicit

---

[1]https://www.festivalnacionalrobotica.pt/2023/en/welcome-en/
[2]https://www.festivalnacionalrobotica.pt/2023/en/autonomous-driving-en/

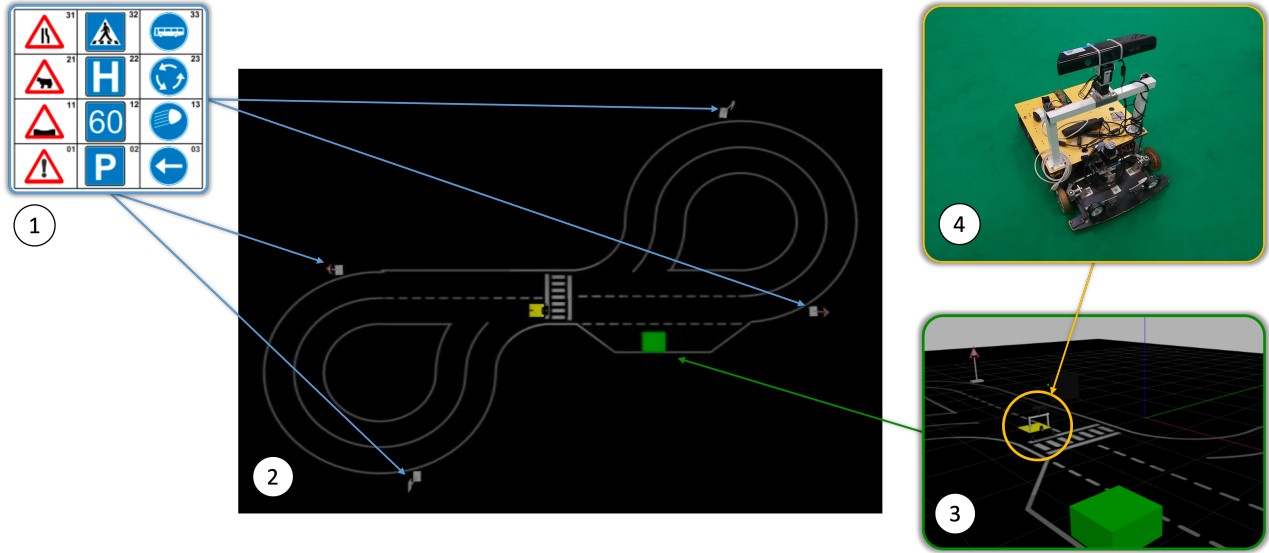

Fig. 2. Overview of the autonomous driving challenge: 1- position in the track of the signs that need to be identified; 2- the track where the robot must conduct multiple laps in an autonomous manner; 3- 3D representation of the track, illustrating the location of an obstacle; 4- ROTA - autonomous tricycle robot.

consideration of motivations) and context scenarios, as presented in what follows. Overall, two personas were considered. Following, their motivations and frustrations associated with the robotics competition are described:

### Persona 1 - Alvin - Researcher

*Motivation*: Alvin is a robotics engineer working at a manufacturer of forest and gardening tools. He is currently developing the new version of Automowers, the company's line of automatic lawn-mowing robots, alongside a small team. Their task is to improve and stabilize the robot's capabilities of obstacle detection, path planning and find new ways of locomotion that can be implemented in the robot, with technologies such as AR, since previous versions had issues when detecting and working around small obstacles, such as flowerbeds or, occasionally, would get stuck while traversing the garden.

*Frustrations*:
- Issues during the competition, such as technical breakdowns that might occur on the robots and this can prevent Alvin from gathering information;
- Alvin might not find adequate information or the kind of data that can be used to help him and his team on their project;
- AR concepts are complex and since Alvin has little to no experience in this field, it might take some time to comprehend them.

### Persona 2 - Henry - Public audience

*Motivation*: Henry is a twelve-year-old boy and the youngest son of Alvin. It's his first time at robot competitions and he has never

interacted with a racing robot before. Henry wants to understand what happens during a competition.

*Frustrations*:
- Issues during the competition, such as technical breakdowns that might occur on the robots;
- Not understanding how to use the application that controls the robot's camera;
- At young age, it may be difficult for him to understand many concepts in this matter.

### Scenarios
To give context on how this work can be proved useful in its desired environment, several scenarios were created, displaying possible uses for the system, being the actors, and the personas presented previously.

*Scenario 1 - Alvin checks his robot logbook:*
As a hobby, Alvin participates in the robotic competition, this time, using the RobARtics tool to check more detailed information, such as the sign identification capabilities of his robot, its orientation while traversing the track, as well as analysis of a heat map to check the accuracy of the robot's path planning methods.

*Scenario 2 - Henry goes to a competition with his father:*
Henry goes with his father to attend his very first robotic competition. Henry is not familiarized with the basic concepts in robotics. As the competition starts, he is able to see additional information about the competing, with the help of the RobARtics tool, which illustrates what is happening.

### Low Fidelity Prototypes

A small prototype was developed using Balsamiq Mokups. The features in the prototype were split in two types of users, the researcher and the general audience, represented by the personas described previously. The goal was to allow them to view additional information on top of the robot through the use of AR.

**Public audience features**

- View Robot Details;
- View Trajectory;
- Enable Sign Identification;
- Enable Ghost Racer.

**Researcher features**

- View Robot Details;
- View Trajectory;
- Enable Sign Identification;
- Enable Ghost Racer;
- View Heat Map;
- View Odometry;
- View Logbook.

The low-fidelity prototypes were tested during a preliminary user study with 6 participants having previous experience with robotics and AR. Participants were instructed to perform a series of given tasks, covering every functionality intended for the application. In the end, the participants were asked to provide feedback.

The first task was to verify the **"Robot Details"**. This page was intended to check the specific information about the registered robots, such as the type of sensors they use and localization techniques, as well as allow them to see how one robot compared to the others (Figure 3-1). Overall, the results collected were positive, being the interface considered intuitive for checking the robot's characteristics.

The second task was to analyze the **"Robot Trajectories"**, which simulated the visualization of the current robot's trajectory or a previous race around the track. From there, the participants were asked to view both trajectory modes. All users considered that this feature was easy to use and that it could be very relevant during the competition.

The following task had participants simulating the **"Sigl identification"** capabilities of a robot. This means that the robot, whenever saw a traffic sign, such as turning on the low beams in a tunnel, or a cattle alert sign would display on the top right of the camera the sign it detected (Figure 3-2). This feature was positively received, as participants found clearly how the prototype demonstrated the robot detecting the sign. This was considered very useful to inform researchers and the audience about what the robot is viewing.

Then, participants had to visualize the **"Ghost Racer"** feature. This simulated a virtual representation of the robot from a previous lap (Figure 3-3), which is useful in comparing the robot's performance between laps. Some participants were familiarized with this concept since it is already used in racing video games. Some of the more inexperienced users said that this feature should remain exclusive to the researcher.

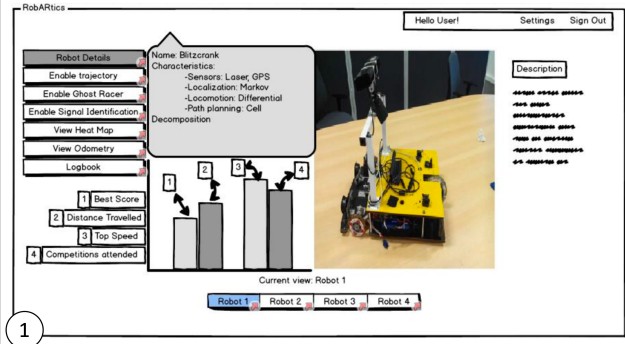

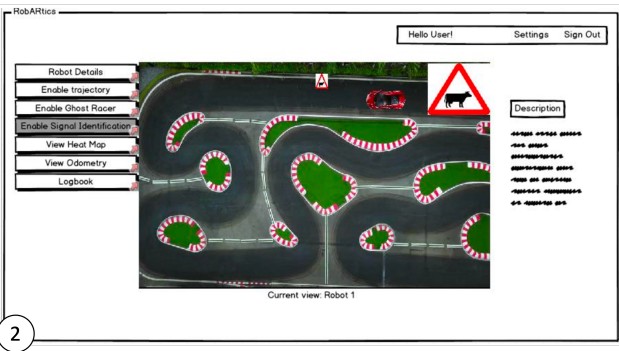

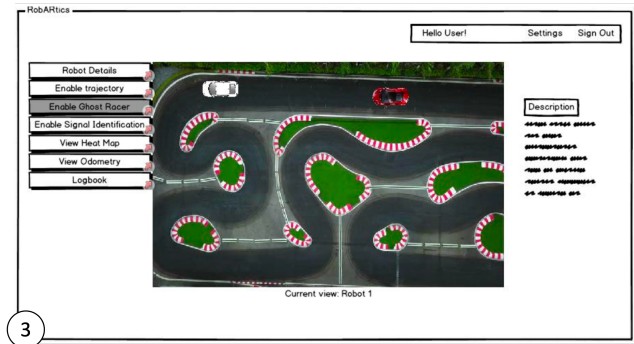

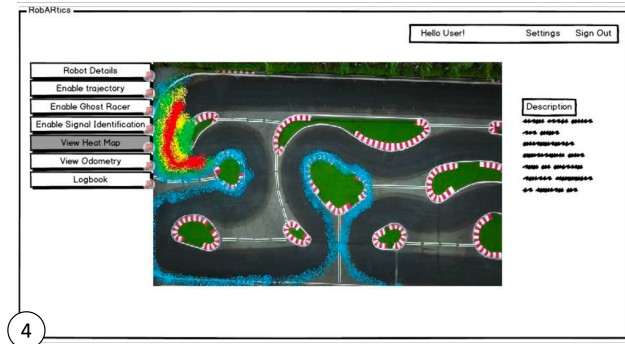

Fig. 3. Low fidelity prototypes: 1- robot details interface; 2- sign identification interface; 3- ghost racer interface; 4- heat map interface.

Next, the task was to observe the heat map generated by the robot. A heat map consists of a visualization technique that uses color to

encode data value (Figure 3-4). In this case, it displays the areas where the robot spent more time during a certain number of laps. Red represents where it stayed the most, followed by yellow, green, and finally blue. Both experienced and novice participants found that this feature should stay exclusive to the researcher, given that it may not be easy to understand by the general audience, which could cause confusion.

Following, the next task was the visualization of the robot **"odometry"**, a method that uses data read from the sensors of the robot to estimate changes in its position over time. Some participants found odometry to be a complex concept to understand and due to this, they could not give proper feedback on this feature. As such, most participants agreed that this feature should not be included in the general audience profile.

Finally, the last task was to check the **"Logbook"**, which contains a history of the recorded values made by the robot, such as road signs detected and the heat map. Generally, participants found this feature interesting, since it allowed them to view all the information and actions performed during the track in a centralized manner.

Despite the positive feedback, some critical suggestions were made, which were integrated during the creation of the prototype described in the next section.

## 3 ROBARTICS TOOL AND INITIAL RESULTS

Following the low-fidelity prototypes and the feedback collected, the RobARtics tool was created to assist the researchers and the general audience during robotic competitions through the use of AR interfaces, which can be displayed on top of a robot (Figure 1).

RobARtics was created with the Unity 3D game engine, based on C# scripts. This engine was chosen for development due to its versatility and ability to develop AR applications, as well as its capacity to communicate with ROS through the ROS# plugin. The virtual content was placed in the real-world environment through the Vuforia library.

Regarding the User Interface (UI) displayed using AR on top of a pre-defined marker (e.g., on top of the ROTA robot), a main screen was created including various components (see Figure 4), namely, a map of the track (on the left side), which is updated according to the information that users intend to visualize, based on a set of check-boxes (bottom of the UI). Besides, detailed information based on the feature selection can be displayed using text and images (on the right side of the screen) to provide additional context that complements what is presented in the map of the track. This way, multiple visualizations can be combined at the same time, unlike the previous design of the low-fidelity prototypes, where each feature was contained in an individual scene, that could only be displayed one at a time. For example, the current trajectory can be displayed, as well as a representation of the signs identified by the robot (see Figure 4-1), according to the list defined by the FNR, allowing to verify if the identification was correct or incorrect quicker than having to check the robot log later on.

Also, distinct colors are used to differentiate various trajectories, e.g., current and previous trajectories (the ongoing trajectory stays on screen and new ones are rebuilt with a new color). To elaborate, the global position, retrieved from the odometry topic, starts to be

drawn on top of the track, represented by several small, connected spheres. While the coordinates are being generated as spheres, they are also being stored in the tool's internal memory, being available to be analyzed if necessary on the logbook (see Figure 4-2).

Another relevant feature is the ghost racer, allowing its users to view and compare various previously saved trajectories, which can be reproduced to check differences in the robot trajectory and possibly try to understand what caused them (see Figure 4-3).

One last feature is the capacity to display additional information regarding the robot, which allows the general audience to have an overview of the robot's characteristics (see Figure 4-4).

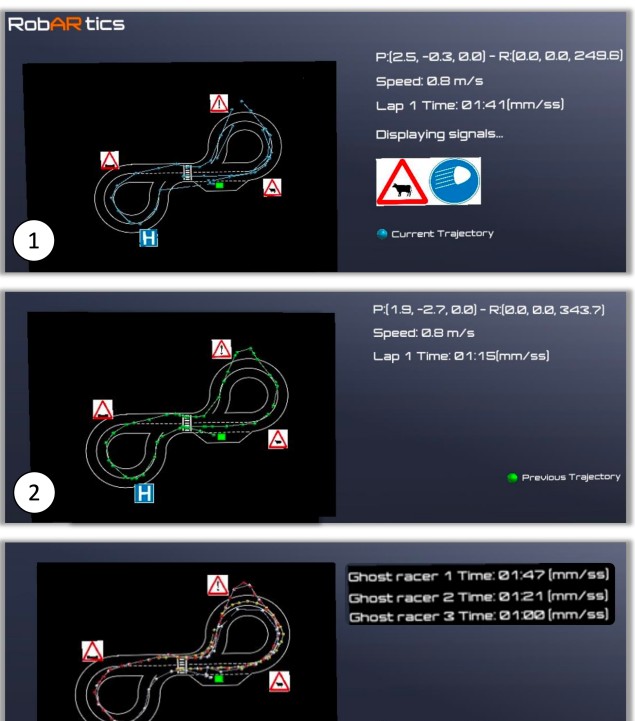

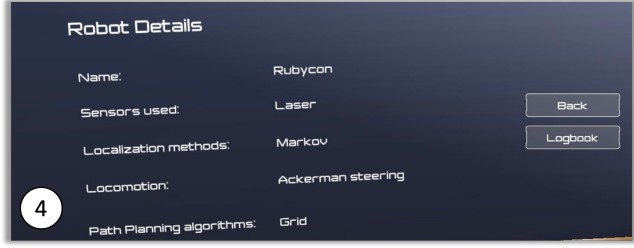

Fig. 4. Example of various AR interfaces displayed on top of the robot in the real-world environment: 1- sign identification; 2- trajectory; 3- ghost racer; 4- robot details.

As before, a user study was conducted to evaluate the user interface of the tool aforementioned. To this end, 5 distinct individuals were recruited (one participant had no background knowledge in

either robotics or AR, two participants had previously interacted in AR environments and the remaining two participants were both familiar with robotics and AR).

Participants were instructed on the setup, the task, and gave their informed consent. Next, they watched a video of ROTA performing some laps around a track, which included an obstacle and road signs located around the track, similar to the competitions this robot is used in. While watching it, participants were given an overview of what they were seeing on the video, the drawbacks of not having any information related to the competition or the robot besides what they were able to directly observe, and the benefits that robARtics could provide during competitions. Then, the AR-based tool was introduced. After finishing the tasks, participants assessed the conditions considered based on the dimensions used. Plus, a small interview also occurred.

Most users understood well what the goals were and generally found the volume of the information displayed to be adequate. All users found that navigating through the tool UI could be done with limited effort, being the information presented clear to understand after some time. Thus, suggesting that the tool might have a small learning curve for new users being introduced to AR, while being easy to pick up for more experienced users. Yet, one participant suggested that the checkboxes used were a bit small, which could affect their responsiveness. Moving away from the UI, a participant suggested the use of a video projector for the general audience, as a means to capture the attention of a larger number of individuals.

After the previous study, the tool was shown to four researchers within the field of robotics that had actually taken part in robotic competitions, particularly in autonomous driving challenges. The goal was to provide more insights into the robARtics tool, including constraints and improvements that can be made. Also, their overall opinions on the introduction of AR within the context of robotic competitions, since they had participated in this kind of events, but with limited interaction with the general audience and reduced support to researchers.

Overall, every researcher understood the application's purpose and what contributions it could give in these environments. Some researchers commended the fact that they believed the marker-based approach would work perfectly, other tracking options could be considered later on. Additionally, they thought that every feature integrated into the robARtics improved or increased the information that could be obtained from the competition itself. One of the researchers said the ghost racer was useful when there was the need to test different path-finding algorithms for the robot, to see which one had the best results in terms of lap time and the robot's on-track accuracy. Other researchers said that the Sign Detection feature was advantageous since some details might escape from the team if they are not always paying attention to their robot, thus not seeing if the robot made the correct sign identification.

Some criticism included the fact that most of the application features were not parameterizable. The researchers expressed that they would like to have control over certain aspects, e.g., select which topics they wanted to subscribe. The former would allow accommodating different track sizes. For example, longer tracks required the position to be updated at a higher rate, so the robot's position is displayed more accurately, while the latter would add

a new layer of usability since researchers could choose to see the position of other robots besides ROTA.

When asked about their thoughts on AR within the context of robotics and how it was integrated with the application, the answers were varied. Two researchers thought that AR was a helpful tool, due to the fact that the tracks which the robots traverse in competitions are of large proportions and they would need to set up one or more cameras to cover the entire track. The application provided a solution to this since they could see the track and the trajectory of the robot by simply pointing a device toward the robot's respective marker. Still, they expressed that they would rather have a desktop version of the application, usable through a webcam since they believed that a mobile device would be better suited for the public watching the competition. The other two researchers shared the same point mentioned previously, as they said that for software development of the robot, a mobile device was not as practical as a computer, since they had to pick it up and point the device towards the robot each time they wanted to see the information.

Conversely, they concluded that the robARtics tool was well thought out, given that the integration of AR provided an innovative and interactive way of clarifying what happens during the robot trials and gave the spectators a means to get more involved in the event. They also found the application to be easily understandable with a bit of practice.

As for feedback and future work improvements, the researchers gave several suggestions: 1- Improve the Sign Identification feature by displaying the real sign and the one detected by the robot side by side, to make the comparison more visible if the robot had not detected a sign in a long time, the icons of the signs it detected previously would be cleaned, to make the interface less cluttered; 2- Add the option to select between an assortment of tracks, representative of other challenges or tracks from previous editions; 3- Add the possibility to track multiple markers, each representative of a different robot, to give users the ability to check information from more robots than ROTA; 4- Expand the AR functionalities by adding the opportunity of inserting virtual obstacles on track for the robot to get around or making the information available proportional to the current zoom (i.e., the displayed information varies the closer or the farther the user is pointing at the marker); 5- Show an augmented representation of the robot's laser, as well as other relevant sensors; 6- Develop an API to make the tool usable by other robots programmed with distinct frameworks besides ROS.

To finish, we intend to apply the robARtics tool during a real-life robotic competition, from which more ecological data can be collected and analyzed to improve the flexibility and suitability of the proposed tool to better support researchers and the general audience moving forward.

## 4 CONCLUDING REMARKS AND FUTURE WORK

This paper described a work that has used an HCD methodology aimed at supporting researchers and the general audience in understanding what happens during a robotic competition through the use of AR technologies. By presenting low-fidelity prototypes to distinct target-users, it was possible to improve the tool design and integrate users feedback during the development phase. Following

this effort, various features were created, namely: view robot details, trajectory, sign identification, ghost racer, heat map, odometry and logbook.

While the investigation conducted sparked interesting results, there are still various research opportunities to be addressed. Moving forward, it could be interesting to also consider the role of the referee, which may also benefit from having additional information on the robots during ambiguous situations which require their intervention. Another interesting topic is to extend the tool range, by supporting desktop devices, which may use a camera to capture the track and provide information to the general audience, e.g., through an external projector. Currently, the tool only allows to visualization of information, but it could also be interesting to integrate methods to allow changing the robot behavior directly from the AR platform. Likewise, enable control of the robot's trajectory using the AR interface, which can facilitate moving the robots in the environment. Finally, we intend to conduct a real-life user study during a robot competition to properly validate the proposed tool with a larger, more diversified audience, a scenario that will provide a more ecological setting.

## ACKNOWLEDGMENTS

To everyone involved in the user studies and discussion sessions, thanks for your time and expertise, in particular Artur Pereira. This research was supported by IEETA, funded through the Foundation for Science and Technology (FCT), in the context of the project [UIDB/00127/2020].

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

Received 20 February 2007; revised 12 March 2009; accepted 5 June 2009

