# OpenReview forum: "RobARtics: Using Augmented Reality to enhance Robotics Competitions"
_humanrobotinteraction.org/HRI/2023/Workshop/VAM-HRI — VAM-HRI 2023 Oral_

### Official Review · Program_Chairs · 2023-02-25
**Accept**

**Rating:** 7
**Confidence:** 5

**Review:**

Review 1:

This paper examines the novel VAM-HRI research area of robot competitions. This exploratory paper begins to explore how AR technology can be utilized to enhance both the experience of the robot operator(s) but the audience as well. The tablet interface provides robot data relating to performance on a race track as well as sub-goals (e.g., object identification). Stakeholders are interviewed a low-fidelity interface is evaluated which strengthens the design process and enhances the analysis of said design. A real AR prototype is also developed and deployed for an initial round of feedback that is used to provide design guidance for future research and iterations of RobARtics. This paper is relevant to the VAM-HRI workshop, and as stated earlier, examines a novel use case of VAM-HRI interfaces that (to my knowledge) has yet to be explored in prior research. I recommend this paper for acceptance.

Questions and Comments:

- The figures do a good job of showing the reader how the interfaces looks and functions.

- Info regarding the hardware used in the interface should be added to the paper (unless I missed it, the only way I knew the interface was on a tablet is by inferring from figure 1).

- I am left wondering how feasible this interface is to audience members if they are required to scan a marker on the robot. I imagine in large robot competitions it would be difficult or impossible for audience members to get that close to competing robots.

- I would be interested in seeing how the 2D tablet interface would compare to a 3D HMD and how this interface could be enhanced to visualize 3D data to the user.

- I think research would be strengthened if there was a baseline to compare against. Since the information displayed on the tablet is displayed on a 2D screen, I question how much the portable nature of the AR interface enhances the user experience (especially since there is no environmentally embedded visualizations).

Review 2:

This paper describes the development of a companion tool, RobARtics, to provide additional information to participants in, or spectators of, robotics competitions. This tool takes the form of a mobile device-deployed AR interface, capable of overlaying relevant information  (robot details, current and prior robot trajectories, etc.) as robots race around a closed course as part of a competition. The authors first introduce a human-centered design approach, taking the perspective of two prospective end-user personas, ranging from a highly knowledgeable participant to a novice spectator. A preliminary study using low-fidelity prototypes was conducted to determine the feature-sets useful to each class of user. Using the feedback obtained, a more validated design was implemented as an AR application, and a series of usability tests were conducted.

Strengths:
- The design process is very thorough - starting with an HCD process, and performing feature-set testing and incorporating feedback before implementation is a valuable cycle.
- This represents a novel use case for information presentation under VAM-HRI, and the future planned user-study evaluating its effectiveness in live robotic competition represents a potentially interesting in-the-wild experiment. I think a few more sentences describing what kinds of data would be collected by such an experiment would be welcome.

Weaknesses:
- The value of this app being implemented in AR is not thoroughly explored as opposed to, say, a standalone mobile application or, as mentioned in your participant feedback, shown on a large projector for the benefit of a large audience. I think more discussion is warranted on why AR specifically is advantageous for this type of tool. There are potential disadvantages of AR related to the information presented being too small to read on a phone, as compared with simply showing the information without AR.
- Relatedly, I think future work should focus on directly comparing the final version of RobARtics with other baseline interfaces.
- My intuition is that the rightmost picture in Fig. 1 is the only image of the RobARtics interface as it actually appears in AR. This should be made clear from the caption, and should be either re-referenced when the AR component is described, or another figure should be added showing the interfaces in their AR context.
- For the camera ready, the paper should conform with the HRI conference paper format to ensure uniformity of presentation. The original submission is in a different format. Additionally, the name of the conference should be entered into the LaTeX, rather than leaving it as “Conference acronym ‘XX, etc.”

Overall, I think this paper would be a good fit for the program at VAM-HRI, and recommend acceptance.

---

### Decision · Program_Chairs · 2023-03-02

Accept (Oral)